# Immunogenicity and Cross-Protection Efficacy of a Genotype F-Derived Attenuated Virus Vaccine Candidate against Mumps Virus in Mice

**DOI:** 10.3390/vaccines12060595

**Published:** 2024-05-30

**Authors:** Seo-Yeon Kim, Hyeran Won, Yun-Ho Hwang, Se-Eun Kim, Jung-Ah Lee, Dokeun Kim, You-Jin Kim, Tae-Young Lee

**Affiliations:** Division of Infectious Disease Vaccine Research, National Institute of Health, Korea Disease Control and Prevention Agency, Cheongju 28160, Republic of Korea; tjdus9284@korea.kr (S.-Y.K.); hw403@korea.kr (H.W.); yunho1129@korea.kr (Y.-H.H.); sekim57@korea.kr (S.-E.K.); jaylee1550@korea.kr (J.-A.L.); dickykim@korea.kr (D.K.); yjiney2@korea.kr (Y.-J.K.)

**Keywords:** mumps virus, attenuated vaccine, immunogenicity, cross-protection

## Abstract

Mumps virus (MuV) causes an acute contagious human disease characterized by swelling of the parotid glands. Despite the near elimination of mumps in many countries, the disease has recurred, even in vaccinated populations, especially adolescents. Immunization effectivity of the genotype A vaccine strain Jeryl Lynn (JL) is declining as genotype A is no longer predominant; therefore, a new vaccine strain and booster vaccine are required. We generated a cell culture-adapted MuV genotype F called F30 and evaluated its immunogenicity and cross-protective activity against diverse genotypes. F30 genome nucleotide sequence determination revealed changes in the NP, L, SH, and HN genes, leading to five amino acid changes compared to a minimally passaged stock (F10). F30 showed delayed growth, smaller plaque size in Vero cells, and lower neurotoxicity in neonatal mice than F10. Furthermore, F30 was immunogenic to other genotypes, including the JL vaccine strain, with higher efficacy than that of JL for homologous and heterologous immunization. Further, F30 exhibited cross-protective immunity against MuV genotypes F and G in *Ifnar*^−/−^ mice after a third immunization with F30 following two doses of JL. Our data suggest that the live-attenuated virus F30 could be an effective booster vaccine to control breakthrough infections and mumps epidemics.

## 1. Introduction

The mumps virus (MuV), belonging to the genus *Orthorubulavirus* of the family *Paramyxoviridae,* has a single-stranded negative-sense RNA genome of approximately 15.3 kb. The genome is arranged, from the 3′ end, in the following order: nucleoprotein (NP), phosphoprotein (P), matrix (M), fusion (F), small hydrophobic (SH), hemagglutinin-neuraminidase (HN), and large (L) proteins [1]. Based on the SH gene sequences, the virus is classified into 12 genotypes, from A to N (except E and M) [2,3]. Mumps is a highly infectious disease characterized by parotid edema caused by MuV infection. Clinical symptoms of MuV infection include fever, headache, myalgia, and parotid gland swelling. MuV is also highly neurotropic, with approximately half of all clinical cases demonstrating its ability to invade the central nervous system (CNS). It leads to meningitis in 10% of cases and encephalitis in less than 1% [4,5,6,7].

Live-attenuated vaccines have proven successful in combating various infectious viruses, such as smallpox, poliovirus, yellow fever, and the measles, mumps, and rubella (MMR) viruses [8]. These vaccines offer several advantages, including a relatively simple manufacturing process, preservation of native antigens, emulation of natural infections, and eliciting a robust immune response [9]. The first live-attenuated mumps vaccine, Jeryl Lynn (JL) B strain, belonging to genotype A, received approval in the USA in 1967. Subsequently, the mumps vaccine became part of the trivalent MMR vaccine. Implementation of the two-dose MMR vaccination program containing the JL strain significantly reduced mumps incidence among school children by the 1990s. Since the introduction of the two-dose MMR vaccine in the Republic of Korea in 1997, there has been a rapid decrease in mumps infections [1,10].

However, despite vaccination efforts, significant mumps outbreaks continue to affect adolescents globally. According to mumps incidence data sourced from the Korea Disease Control and Prevention Agency’s infectious disease, available on 2 February 2024 (https://dportal.kdca.go.kr/pot/) spanning the decade from 2012 to 2022, over 15,000 mumps cases have been consistently reported annually since 2013. Notably, in 2014, there was a significant surge in mumps infections among adolescents aged 13–18 years, with reported cases reaching 13,603.

The resurgence of mumps outbreaks can be attributed to waning immunity and antigenic differences between vaccine strains and circulating strains. Findings from the “2014 Measles, Mumps, and Rubella Immunity Survey” conducted domestically revealed that the antibody positivity rate was 74% among individuals aged 2–3 years, 86% among those aged 4–6 years, and 89% among those aged 7–9 years. However, this rate decreased to 62% among individuals aged 13–18 years. Additionally, immunity surveys conducted overseas have reported a significant decline in antibody levels and disparities in the avidity of mumps immunoglobulin G (IgG) over time post-vaccination, compared to measles and rubella [11,12]. Therefore, waning immunity post-vaccination emerges as a plausible explanation for the increased incidence of mumps despite high vaccination rates. Hence, additional vaccination is recommended for children and adolescents.

The F, H, and I MuV genotypes are prevalent in East Asian countries, and the G genotype is prevalent in Europe and the United States [13,14]. In Korea, the F genotype was briefly prevalent in 2008, and the H and I genotypes have been largely prevalent since 2007. It has been reported that the commercial genotype A vaccine strains (JL and RIT 4385) are genetically different from epidemic strains and that there may also be differences in immunogenicity [9,10,15,16,17]. It has been predicted that a decrease in cross-protection efficacy due to differences in MuV genotypes causes breakthrough infections. Therefore, there is a need to develop a pandemic-ready booster vaccine that responds to various genotypes and induces a high neutralizing ability. Previous studies confirmed that the mumps F genotype has cross-protective effects against various genotypes [18]. In this study, we developed a new attenuated MuV vaccine candidate by serially passaging the F genotype in Vero cells. We investigated the characteristics of the newly attenuated MuV vaccine candidate and evaluated its immunogenicity and protective ability in mice.

## 2. Materials and Methods

### 2.1. Cells and Viruses

Vero cells were purchased from the American Type Culture Collection (ATCC; Cat#: CRL-81) and cultured in Dulbecco’s modified Eagle’s medium (DMEM; Gibco, Grand island, NY, USA) supplemented with 10% fetal bovine serum (FBS) and 1% penicillin-streptomycin (P/S) at 37 °C in 5% CO_2_. The JL, an attenuated genotype A mumps viral vaccine strain, was obtained from the Korean Ministry of Food and Drug Safety (No. 0666296). MuVi/Incheon.KOR/16.08/22[F] (F genotype), MuVi/Incheon.KOR/09.15/1[H] (H genotype), and MuVi/Jeonnam.KOR/10.15/5[I] (I genotype) were isolated from the throat and saliva of patients, and MuVi/Iowa.US/2006[G] (G genotype) was obtained from the ATCC (VR-1899). Vero cells were inoculated with the viruses in infection media [minimum essential medium (MEM) containing 2% (*v*/*v*) FBS and 1% (*w*/*v*) P/S]. The virus-infected cells were incubated at 37 °C in 5% CO_2_ and harvested when the virus-specific cytopathic effect reached 70–80%, and then they were stored in aliquots at −80 °C.

### 2.2. Generation of an Attenuated Genotype F-Based MuV Vaccine Candidate

Vero cells were seeded in 75T flasks and incubated until they reached approximately 90–100% confluency. Minimally passaged MuV (F genotype, MuVi/Incheon.KOR/16.08/22) was inoculated into the Vero cells and incubated by gently shaking the plates several times for 1 h. The inoculum was removed and replaced with 10 mL fresh infection medium. After 3 days incubation, virus-infected Vero cells were quickly frozen and stored at −80 °C. After thawing, the cells were harvested, and the viral supernatant was stored in aliquots. This process was repeated approximately 20 times and the final virus F30 was used in the experiment in a mass-produced, concentrated, and purified form.

### 2.3. Viral RNA Isolation and RNA Deep Sequencing

RNA was extracted from frozen viral samples to produce an RNA library using the QIAseq FX Single Cell RNA Library Kit (Qiagen, Hilden, Germany). The library was transferred to the flow cell to create clonal clusters. After the clusters were formed, the flow cells were installed in the next generation sequencing equipment (Illumina Hiseq 4000; Illumina Inc., San Diego, CA, USA) to perform RNA sequencing, and the sequences were obtained through a series of processes. After quality control, the raw sequence data were assembled using gsMapper (V2.8) to produce contigs, and using these contig sequences, the full-length sequence was completed using Proovread (V2.14.1). The whole-genome sequences of the pre- and post-attenuated viruses were analyzed using the Benchling platform and BioEdit v.7.2.5 Alignment program.

### 2.4. Growth Kinetics of Viruses in Vero Cells

Cultures of Vero cells in 75T flasks were infected with JL, a minimally passaged F genotype virus (F10), and a cell-culture-adapted F genotype (F30) at a multiplicity of infection of 0.01. A small volume of each inoculum was collected every 24 h for 4 days after infection, and the viral titer was determined using a plaque assay. Vero cells were seeded at a density of 1 × 10^5^ cells/mL in 24-well plates containing DMEM supplemented with 10% (*v*/*v*) FBS and 1% (*w*/*v*) P/S. When the cells reached approximately 90–100% confluency, serial dilutions of MuV were prepared and the Vero cells were inoculated and incubated at 5% CO_2_, 37 °C for 1 h. The inoculum was removed and replaced with an overlay medium [1.5% (*w*/*v*) carboxymethylcellulose in MEM containing 2% (*v*/*v*) FBS and 1% (*w*/*v*) P/S]. The cells were incubated for a further 6 days and stained with crystal violet solution for 1 h. Plaques were counted in each well, and viral titers were calculated. To determine viral plaque sizes, the plate bottoms of the samples harvested at 48 h were scanned, and the plaque areas were measured using ImageJ v.1.48 software. Ten plaques were randomly selected, and the mean area was calculated as the final size of the plaques.

### 2.5. Neurotoxicity Test in Neonate C57BL/6 Mice

A newborn mouse model was used for the neurovirulence assessment. One-day-old suckling C57BL/6 mice (*n* = 4–6) were inoculated with 10^4^ and 10^5^ tissue culture infective doses (TCID_50_) of the F10 and F30 viruses by intracerebral (i.c) injection in a 10 µL volume using a 31-gauge needle as previously described [19]. The inoculation site was located in the right parietal area of the skull approximately 1 mm to the right of the midline. On day 25 after inoculation, the mice were anesthetized by appropriate doses of Rompun and Ketamine and euthanized with CO_2_ gas after sample collection as recommended. The brains were aseptically removed and fixed in 10% buffered formalin. Three parts of the brain tissue were cut horizontally from the front to an appropriate size and embedded in paraffin. After paraffin-treated blocks were made, the tissues were sectioned to a thickness of 3 µm and stained with hematoxylin and eosin. Stained tissues were observed under an optical microscope (Olympus, Tokyo, Japan). The number of lesions observed in the three brain sections was counted and scored depending on the degree of severity.

### 2.6. Immunogenicity and Protection Test in Mice

For the homologous vaccination test, C57BL/6 mice (female, 4-week-old) were used for immunogenic analysis. Mice were immunized at 2-week intervals via the intramuscular (i.m.) route with MuV vaccines at 1 × 10^5^ plaque-forming units (PFUs) or mock. The same mice were used for booster immunity tests. The first booster inoculation was performed 2 weeks after priming, and the second booster was performed 8 weeks after the first booster. Serum was collected 14 days after the last vaccination for enzyme-linked immunosorbent assay (ELISA) and neutralization assay, and splenocytes were isolated from mice for enzyme-linked immunospot (ELISpot) assays.

For the protective immunity test, interferon α/β receptor knockout (*Ifnar*^−/−^) mice [B6(Cg)-Ifnar1.2Ees/J: JAX-M-2492] were used as animal models for MuV challenge [20]. The immunization conditions mirrored those outlined for booster vaccination. Two weeks after the delayed immunization, transgenic mice were intranasally challenged with 30 µL of phosphate-buffered saline (PBS) containing 3 × 10^6^ PFUs of either the F or G genotype MuV. After viral infection, the mice were euthanized on the second, fourth, and seventh day to measure the viral titers in the lung tissues.

#### 2.6.1. ELISA

MuV antigens were produced by infecting Vero cells with genotype A, F, H, I, and G MuVs. Immunolon^®^ Microtiter^TM^ high binding polystyrene plates (Thermo Scientific, Waltham, MA, USA) were coated with each of the five genotypes of MuV at 5 × 10^4^ PFUs/well overnight at 4 °C. The plates were washed and blocked with 200 µL of PBS containing 5% skim milk for 1 h at 25 °C. Serum was serially diluted (three-fold) starting at 1:100 in PBS containing 3% skim milk and added to each virus-coated well. Horseradish peroxidase-conjugated goat anti-mouse IgG secondary antibody (Abcam Cat#: AB6789) was used at a dilution of 1:5000. The plates were developed using a TMB substrate solution (GeneDepot, Katy, TX, USA). The absorbance at 450 nm was measured using an ELISA reader (SpectraMax i3x; Molecular Device, San Jose, CA, USA) and the dilution factor at the cutoff value was calculated as the ELISA titer. The cutoff value was calculated by the mean absorbance at 450 nm of the negative control.

#### 2.6.2. Plaque Reduction Neutralization Test (PRNT)

Mouse serum samples were heat-inactivated at 56 °C for 30 min. Serum samples were prepared as a 2-fold dilution series in infection media, starting at 1:10. Serially diluted sera were added to an equal volume of diluted virus (50 PFUs per well) and incubated for 1 h. After incubation, cells at approximately full confluence in 24-well plates were infected with the virus–serum mixture for 1 h. The virus inoculum was gently removed from each well, and the wells were covered with 0.5 mL per well of overlay media and incubated for 6 days. After fixing and staining the cells, 50% neutralizing antibody titers were calculated using the modified Kaber formula as described previously [21].

#### 2.6.3. Interferon-Gamma ELISpot Assay

ELISpot was performed using the Mouse IFN-γ ELISpot Plus Kit (MABtech, Nacta Strand, Sweden) following the manufacturer’s protocol. Spleens were removed from the mice, and cells were isolated by mechanical disruption using a gentleMAX machine (Miltenyl Biotec, Bergisch Gladbach, Germany). After filtering with 100 μm pore size strainers, red blood cells were removed using an ACK lysis buffer (Lonza, Basel, Switzerland). The cells were suspended in RPMI containing 10% FBS and 1% P/S and seeded at 5 × 10^5^ cells per 100 µL. The cells were stimulated with PMA/ionomycin, 1 µg/mL of 0.05% formalin-inactivated F and JL viruses, or media for 12–48 h. Plates were developed using R4-6A2-biotin and streptavidin-ALP antibodies. A ready-to-use substrate solution was used for spot emergence, and the spots were visualized using CTL ImmunoSpot equipment (S6 Universal M2 v.7.0; Immunospot, Cleveland, OH, USA).

#### 2.6.4. Viral Load Detection in Infected Lung Tissue by Quantitative Real-Time Reverse-Transcription Polymerase Chain Reaction (qRT-PCR)

After the third immunization of *Ifnar*^−/−^ mice, lung tissues were isolated on days 2, 4, and 7 after challenge with genotypes F or G. The lung tissues were homogenized using the Precellys CK28 Lysing Kit (Bertin, Paris, France; Cat#: P000911-LYSK0-A.0), and the residual viral RNA was extracted using a Maxwell RSC 48 Instrument (Promega, Madison, WI, USA; Cat#: AS8500) and a Maxwell RSC Viral Total Nucleic Acid Purification Kit (Promega, Madison, WI, USA; Cat#: AS1330) according to the manufacturer’s protocols. The virus remaining in the lung tissue was quantified by qPCR using a PowerChek^TM^ Mumps Virus Real-time PCR Kit Ver.1.0 (Kogene biotech, Seoul, Republic of Korea; Cat#: R3110C) to measure the protective immunity of the vaccine candidate.

### 2.7. Statistical Analysis

Data were analyzed and graphs plotted using Prism v.9 (GraphPad Software Inc., San Diego, CA, USA). Data for grouped pairs were analyzed using a two-way analysis of variance (ANOVA) test and expressed as the standard deviation of independent experiments. Statistical significance was set at *p* < 0.05.

## 3. Results

### 3.1. Generation of F30 and Comparison of Genome Changes within F30 and the Reference Viruses

To generate a live-attenuated virus, the minimally passaged mumps genotype F virus (F10) was passaged approximately 20 times in Vero cells and designated F30. Sequential alignment was performed using whole-genome sequences of each virus to investigate the changes in the genome between the parent F0 (GenBank sequence No. MN630056.1), F10, and F30 viruses. There were 14 nucleotide changes between F0 and F30: four mutations in the L protein; two in the NP, F, and HN proteins; and one in the SH protein. One insertion nucleotide and two substitution mutations were identified in the untranslated region (UTR) as shown in Table 1. Among them, nine nucleotide mutations led to amino acid changes in F30 compared to F0: three amino acid changes in the L protein, two changes in the F and HN proteins, and one change in the NP and SH proteins. The positions of the mutations in the amino acid sequence are as follows: Ala→Thr^120^ in the NP; Asp→Asn^78^ and Met→269Val^269^ in the F protein; Leu→Pro^57^ in the SH protein; Thr→Ala^154^ and His→Asn^498^ in the HN protein; and His→Asn^818^, Lys→Arg^1406^, and Pro→Gln^1946^ in the L protein. Compared to F10, there were 10 nucleotide changes in the F30 genome leading to five amino acid changes in the NP, SH, HN, and L proteins.

### 3.2. Growth Properties of Attenuated F30 Virus in Vero Cells

We evaluated the replication kinetics of the JL, cell culture-adapted F10, and F30 viruses in Vero cells at a multiplicity of infection (MOI) of 0.01. The virus growth of F30 was delayed by 1 day compared with the JL and F10. The peak titer of F30 was 6.32 log_10_ PFU/mL, which was lower than that of JL and F10 (7.46 log_10_ PFU/mL and 7.30 log_10_ PFU/mL) at 3 days after infection. We also assessed the morphological changes in the plaque phenotype by attenuation. The plaques produced by F10 were larger than those produced by F30. When the sizes of 10 random plaques were measured, those of the attenuated viruses JL and F30 were significantly smaller than the wild-type F10 (Figure 1B,C). The plaques of the F10 virus had a clear edge, transparent inner part, and constant size; however, the plaque shape of the F30 virus was unclear and blurry, and the sizes of the plaques were irregular.

### 3.3. Neurotoxicity Test of F30 in Neonate C57BL/6 Mice

The neurotoxicity is a major concern for the development of new mumps vaccine candidates. To evaluate the neurovirulence of F30 in vivo, newborn C57BL/6 mice were intracerebrally inoculated with PBS, JL, F10, or F30 viruses. At 25 dpi, the mice were euthanized, and the severity of inflammatory lesions was determined by grading the score of microglia, perivascular cuffing, and leptomeningitis in the brain as previously described [22]. All findings were pathological diagnoses related to inflammation, which were mainly observed in the middle and back of the brain. Representative cross-sections for each viral infection are shown in Figure 2A. MuV-induced hydrocephalus, which is a major neuropathological outcome of MuV infection, was also assessed. No specific abnormalities were observed in any mice in the PBS group. Surprisingly, we found that F30 induced relatively lower neurotoxicity and inflammatory responses than JL and F10. In the JL and F10 group, there was a correlation between the occurrence and degree of inflammation-related findings; however, in the F30 group, no correlation was detected (Figure 2B).

### 3.4. Humoral and Cellular Immunity in C57BL/6 Mice Immunized with F30

To explore the immunogenicity of the F30 attenuated vaccine candidate, 4-week-old C57BL/6 mice were immunized twice with F30 or JL as a positive control at 2-week intervals and serum was collected at termination (Figure 3A). Another group of mice were immunized with PBS as a negative control. MuV-specific serum IgG titers were measured using ELISA. The OD values in accordance with serum dilution are illustrated in Figure 3B, and the titers were determined as the serum dilution with an OD of 0.5, which was the mean absorbance at 450 nm of the PBS-immunized samples, for each sample (Figure 3C). The ELISA titers showed a similar pattern for both JL and F30; however, F30 showed a higher antibody titer than JL at the same dilution point.

To measure the neutralizing antibody (NAb) titer in the serum after two doses of JL or F30, a PRNT assay was performed against genotype A, F, H, I, and G viruses. F30-immunized mice had higher cross-protective NAb titers than JL-immunized mice against all genotypes, except genotype A, even though it is an F genotype-based attenuated vaccine. NAb titers of the control sera from the PBS-immunized mice yielded background levels of inhibition (Figure 3D).

To evaluate the cellular immunity, splenocytes from immunized mice were stimulated in vitro with inactivated JL or F30 viruses and the number of T cells that express IFN-γ were measured by ELISpot assays. The splenocytes effectively produced a greater quantity of IFN-γ in the F30-immunized group than in the JL-immunized group when stimulated with either JL or F30 inactivated viruses (Figure 3E).

### 3.5. Immunogenicity of F30 as a Third Dose Immunization in C57BL/6 Mice

Additional evaluation of heterologous immunization was performed to determine whether F30 could complement waning immunity, which is a limitation of the MMR vaccine. The first and second immunizations were conducted with JL, and 8 weeks later, the third immunization was performed with JL or F30 (Figure 4A). Serum samples were collected for measurement of NAb titers against MuV (genotypes A, F, H, I, and G) and splenocytes were isolated for INF-γ ELISpot 14 days after the last immunization.

The mean NAb titers of the group vaccinated with two doses of JL and one dose of F30 exhibited slightly higher titers against all genotypes than the group vaccinated with three doses of JL (Figure 4B).

In an experiment to examine the T cell immune response through IFN-γ ELISpot assays, the IFN-γ levels were comparable between the F30-third-immunized group and the JL-third-immunized mice stimulated with both inactivated MuV JL and genotype F (Figure 4C).

### 3.6. Cross-Protection Efficacy of F30-Third-Immunized Ifnar^−/−^ Mice against Circulating Genotypes F and G

Interferon α/β receptor knockout (*Ifnar*^−/−^) mice, a common model for in vivo studies of viral infections, were susceptible to MuV and virus titers in the lungs were detected [15]. To determine the protective efficacy of additional vaccination with F30, *Ifnar*^−/−^ mice were immunized with two doses of JL at 2-week intervals and then immunized with JL or F30 at 10-weeks post-prime. After the second boosting, *Ifnar*^−/−^mice were intranasally challenged with the domestic circulating genotype F and the foreign genotype G MuVs, and viral loads in the lungs were measured at 2, 4, and 7 dpi by qRT-PCR (Figure 5A). The lowest gene copy number was at 7 dpi, and the viral loads significantly decreased in the F rather than the G genotype-infected mice. In both groups challenged with genotypes F and G, viral loads in the lungs from JL- or F30-third-immunized mice were significantly lower than those in PBS-immunized mice (Figure 5B,C). There was no difference in the protective efficacy against MuV between the JL- and F30-third-immunized groups, probably due to the lack of a complete immune response in *Ifnar*^−/−^ mice. We compared the production of NAbs after the third vaccination with JL and F30. NAb titers in the F30-third immunized group against genotypes A and I were akin to those in the JL-third-immunized group. However, the F30-third vaccinated group exhibited higher antibody titers against circulating genotypes F, H, and G compared to the JL-immunized group. While both the F30- and JL-third vaccination groups demonstrated similar protective effects against genotypes F and G in *Ifnar*^−/−^ mice, these findings suggest that F30 could serve as an alternative booster vaccine candidate to curb the resurgence of mumps, considering the prevalence of current circulating genotypes.

Interferon α/β receptor knockout (*Ifnar*^−/−^) mice, a common model for in vivo studies of viral infections, were susceptible to MuV and virus titers in the lungs were detected [15]. To determine the protective efficacy of additional vaccination with F30, *Ifnar*^−/−^ mice were immunized with two doses of JL at 2-week intervals and then immunized with JL or F30 at 10-weeks post-prime. After the second boosting, *Ifnar*^−/−^mice were intranasally challenged with the domestic circulating genotype F and the foreign genotype G MuVs, and viral loads in the lungs were measured at 2, 4, and 7 dpi by qRT-PCR (Figure 5A). The lowest gene copy number was at 7 dpi, and the viral loads significantly decreased in the F rather than the G genotype-infected mice. In both groups challenged with genotypes F and G, viral loads in the lungs from JL- or F30-third-immunized mice were significantly lower than those in PBS-immunized mice (Figure 5B,C). There was no difference in the protective efficacy against MuV between the JL- and F30-third-immunized groups, probably due to the lack of a complete immune response in *Ifnar*^−/−^ mice. We compared the production of NAbs after the third vaccination with JL and F30. NAb titers in the F30-third immunized group against genotypes A and I were akin to those in the JL-third-immunized group. However, the F30-third vaccinated group exhibited higher antibody titers against circulating genotypes F, H, and G compared to the JL-immunized group. While both the F30- and JL-third vaccination groups demonstrated similar protective effects against genotypes F and G in *Ifnar*^−/−^ mice, these findings suggest that F30 could serve as an alternative booster vaccine candidate to curb the resurgence of mumps, considering the prevalence of current circulating genotypes.

## 4. Discussion

Mumps is an important vaccine-preventable childhood disease caused by the MuV. The resurgence of mumps has been frequently reported worldwide in recent years, especially in countries that immunize against mumps [1,2,3,4,5,6,7,8,9]. This has resulted in a call for the development of new and more effective mumps vaccines. The vaccine strain JL belongs to genotype A, which is no longer a circulating genotype, whereas genotypes F, G, H, and I currently are.

Live-attenuated vaccines are live viruses weakened by the deletion or mutation of virulence components in the viral genome. They preserve the native conformation of viral antigens and present antigens to the immune system, similar to natural infections. Live-attenuated virus vaccines are among the most immunogenic vaccines and have a long history of success in preventing a variety of infectious diseases, such as polio (Sabin), measles, mumps, and rubella.

In the present study, we generated a new attenuated live virus vaccine candidate by the serial passaging of MuV genotype F in Vero cells and tested its immunogenicity and cross-protection capabilities against domestic and overseas epidemic strains when used as a booster vaccine in mice. We previously reported that serum from inactivated genotype F virus-immunized mice has a cross-protective ability against various genotypes [16,18]. Thus, a new attenuated vaccine using the same platform as MMR was developed. A cell culture-adapted stock, F30, was obtained after 20 passages in Vero cells of a minimally passaged MuV genotype F stock (F10).

In comparing the genome sequences of the F10 and F30 strains, nine nucleotide changes were found in the UTR and coding regions. Five of the changes led to amino acid substitutions: one each in NP (Ala→Thr^120^), SH (Leu→Pro^57^), and HN (Thr→Ala^154^), and two in L (His→Asn^818^ and Pro→Gln^1946^). F30 showed significant attenuation of replication in Vero cells, and the morphology of the plaque changed to a small and irregular shape, similar to that of JL.

Mumps is a highly neurotrophic disease. In the past, mumps vaccines other than JL were discontinued due to central nervous system complications [13,23]. Ferrets and monkeys are susceptible to MuV infection [5,24], However, their models are expensive and difficult to manage. Additionally, some vaccines that appeared neuroattenuated (weakened) in monkeys were later found to be neurovirulent (pathogenic to the nervous system) in humans. This highlights the limitations of using small numbers of monkeys in safety testing. Small animal models, such as rats and mice, offer a better alternative because they are easier to handle and breed. They also allow for larger study sizes, which facilitates obtaining statistically significant results in research. Studies have shown that newborn rats can be used to evaluate the neurotoxicity of mumps vaccines by assessing the severity of hydrocephalus (fluid buildup in the brain) following intracerebral (in the brain) immunization with the mumps virus [25,26]. While adult mice are resistant to MuV infection, newborn mice infected with the virus develop lung lesions and show detectable viral titers in the brain [27,28,29]. Therefore, we employed a newborn mouse model to evaluate the neurotoxicity of our vaccine candidate. Our findings indicate that the F30 strain exhibited lower neurovirulence in neonatal mice compared to the JL and pre-attenuated F10 strains. Notably, the neurotoxicity of the JL strain was comparable to that of the F10 strain.

L and NP are involved in viral replication and transcription. L forms a complex with P, which acts as a replicase to synthesize new viral genomes and as a transcription factor to generate mRNAs. NPs function in viral RNA encapsulation and RNP complex formation. HN facilitates viral attachment and internalization by binding to the cellular receptor sialic acid. SH is dispensable for viral growth in cultured cells; however, in vivo studies suggest that SH plays a role in viral pathogenesis [23,30]. Similar observations have been reported that amino acid changes in the NP, L, HN, F, and SH proteins are associated with changes in neurotoxicity; for example, Ser→Asn^466^ and Lys→Arg^335^ in the HN protein, Ala/Thr→Thr^91^ and Ser→Phe^195^ in the F protein, and Ile→Val^736^ and Glu→Asp^1165^ in the L protein [31,32,33,34]. Recombinant MuV(S79) carrying mutations in the L protein leads to delayed viral growth in cell culture and the formation of significantly smaller plaques compared to the parental virus. Despite the attenuated viral growth, cotton rats vaccinated with these rMuV mutants produced high levels of NAbs and were completely protected against challenge with a genotype F MuV [7]. Our results, together with those of previous studies, suggest that changes in the expression of these proteins are related to the observed in vitro and in vivo findings of decreased growth and neurovirulence of the F30 strain. The same amino acid substitutions were not observed in the examples mentioned above; evidence suggests that the HN and L proteins are associated with neuroattenuation in newborn mice and exhibit attenuated growth and alterations in plaque morphology in Vero cell culture. Whether changes in these amino acids contribute to viral growth and neurovirulence must be addressed using a reverse genetics system.

The pro-inflammatory cytokine IFN-γ is a good indicator of T-cell immunity. Our results suggest that the induced T-cell immune responses by the F30 vaccine prevented the spread of MuV. The mice immunized with two doses of F30 produced higher NAb than the JL-immunized mice against all tested genotypes, except A. Serum from JL-immunized mice neutralized genotype A more effectively than genotypes F, H, I, and G, which could lead to reduced cross-protective activity between the vaccine and circulating genotypes. After two doses of JL immunization, the F30-third-immunized mice generated a higher titer of NAb than the JL-third-immunized mice against all tested genotypes, including A. We also compared the cross-protective efficacy in *Ifnar*^−/−^ mice immunized with three doses of JL or two doses JL and one dose of F30. In mice challenged with genotype F or G, viral genomes in the lungs were reduced in both the group immunized with the third dose as JL or as F30, compared to those in PBS-immunized mice. Although there was no difference in cross-protective activity against the circulating G and F genotypes between the mice immunized with JL or F30 as the third dose, the challenge MuV gene copy number was significantly reduced compared to infected mice with PBS only, the mock immunization. *Ifnar*^−/−^ mice, a common model for in vivo studies of many viral infections, are susceptible to MuV and are used as an animal model for evaluating protective immunity against MuV. Concurrently, the *Ifnar^–/–^* mouse model has shortcomings due to the lack of the initial antiviral response mediated by type I IFNs; therefore, the animals cannot effectually induce a complete immune response to vaccination. This may explain the lack of difference between the responses to the third immunization with JL and F30 against the MuV. A previous study also reported that the passive transfer of serum from immunized mice protected against MuV infection in *Ifnar*^−/−^ mice. Passive transfer of serum from naïve mice after the third immunization with JL or F30 to *Ifnar*^−/−^ mice infected with different genotypes could strengthen our findings regarding the cross-protective efficacy of F30.

In this study, we generated a live genotype F-based attenuated virus vaccine candidate and assessed its immunogenicity and cross-protection efficacy against currently circulating genotypes. F30 elicited both neutralizing antibodies and cell-mediated immune responses against the MuV in immunized mice. Furthermore, F30 showed low neurotoxicity in newborn mice and cross-protective activity against the F and G genotypes in *Ifnar*^−/−^ mice immunized with two doses of JL and one dose of F30. Based on these results, we hypothesize that a third dose, later in life, with a genotype F-derived F30 vaccine may be beneficial in preventing further resurgence of mumps.

## Figures and Tables

**Figure 1 vaccines-12-00595-f001:**
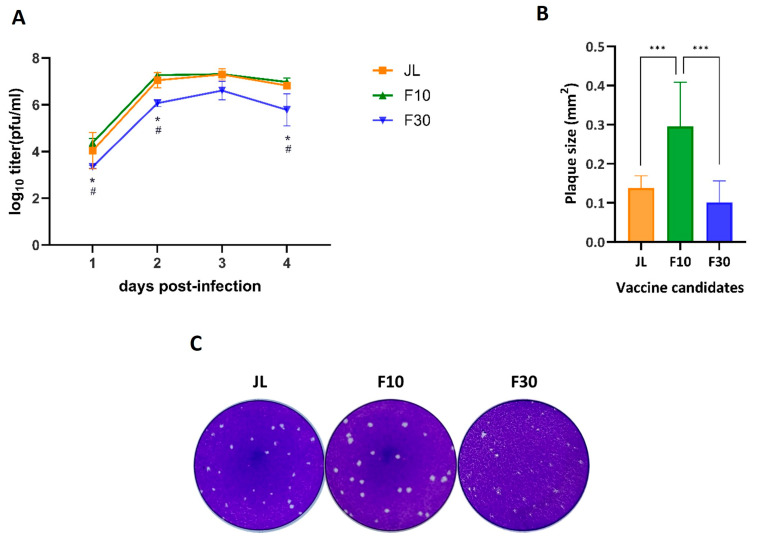
Characterization of cell culture-adapted viruses. Vero cells were infected at an MOI of 0.01. (**A**) Growth curves for the JL, F10, and F30 viruses. Viral titers were determined in the culture supernatants at the indicated times post-infection using plaque-forming assays in duplicate. The results are representative of three independent experiments. Statistical analysis was performed by two-way ANOVA and Tukey’s multiple comparison; * *p* < 0.05 vs. PBS group; # *p* < 0.05 vs. JL group. (**B**) Comparison of plaque sizes. Virus-infected cells at 2 dpi were stained with crystal violet, and the sizes of 10 plaques measured using ImageJ software. The error bars on the graph indicate the standard deviation. One-way ANOVA and Tukey’s multiple comparison test were performed; *** *p* < 0.001. (**C**) Plaque morphology of Vero cells. The plaques formed by the indicated viruses at 6 dpi were stained with crystal violet.

**Figure 2 vaccines-12-00595-f002:**
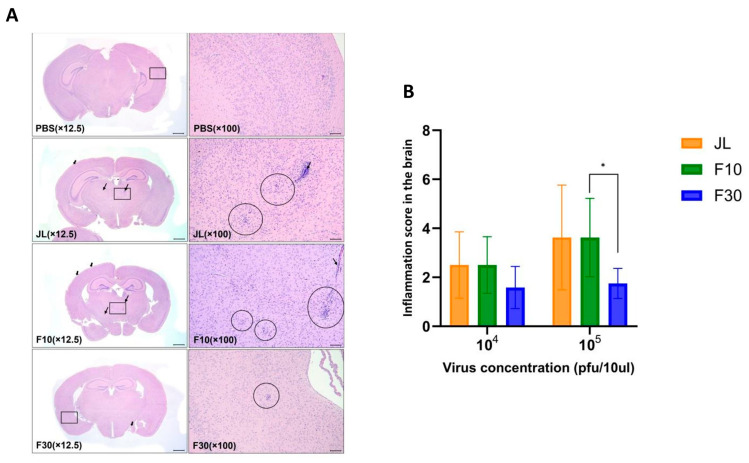
Neurotoxicity of vaccine candidates in neonate mice. (**A**) Representative histological images of hematoxylin and eosin-stained brain tissue sections of infected mice. The mouse brains show microgliosis (circles), perivascular cuffing (thin arrows), and meningitis (thick arrows). The square box represents a 100× magnified part. Scale bars = 800 µm for 12.5× and 100 µm for 100× magnifications. (**B**) Measurement of the severity of hydrocephalus in neonate mice inoculated with the JL, F10, and F30 viruses. (*n* = 4–6; two-way ANOVA and Sidak’s multiple comparison test were performed; * *p* < 0.05).

**Figure 3 vaccines-12-00595-f003:**
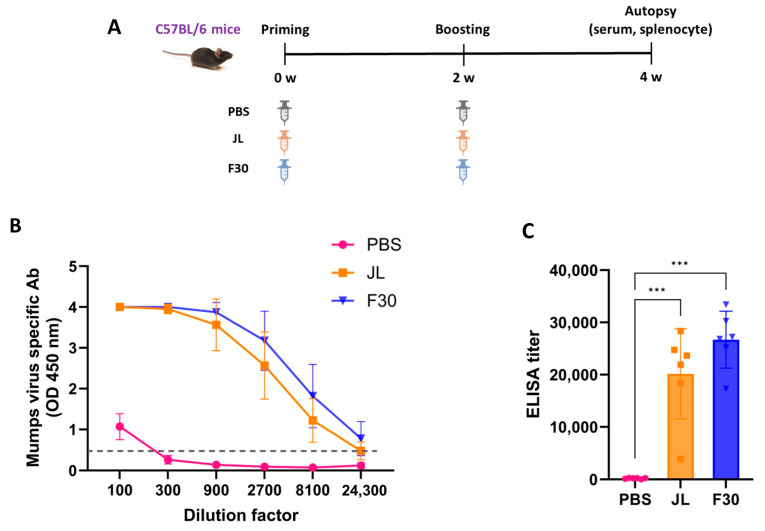
Humoral and cellular immunity in C57BL/6 mice immunized with F30. (**A**) Experimental schematic for homologous immunization. Mice were immunized with two doses of phosphate buffered saline, JL or F30 (*n* = 6). (**B**,**C**) Kinetics of mumps-specific antibody responses against an inactivated MuV mixture of genotypes A, F, H, I, and G measured by ELISA. One-way ANOVA and Tukey’s multiple comparison test were performed; *** *p* < 0.001. (**D**) Neutralizing antibody titers against several genotypes of mumps virus measured by PRNT assays. Two-way ANOVA and Tukey’s multiple comparison test were performed; * *p* < 0.05, ** *p* < 0.01, *** *p* < 0.001 vs. PBS group; ## *p* < 0.01, ### *p* < 0.001 vs. JL group. (**E**) T cell responses measured by IFN-γ ELISpot assays. Splenocytes from immunized mice were isolated and stimulated with inactivated F30 and JL viruses. Two-way ANOVA and Tukey’s multiple comparison test were performed; ** *p* < 0.01, *** *p* < 0.001 vs. PBS group; ### *p* < 0.001 vs. JL group.

**Figure 4 vaccines-12-00595-f004:**
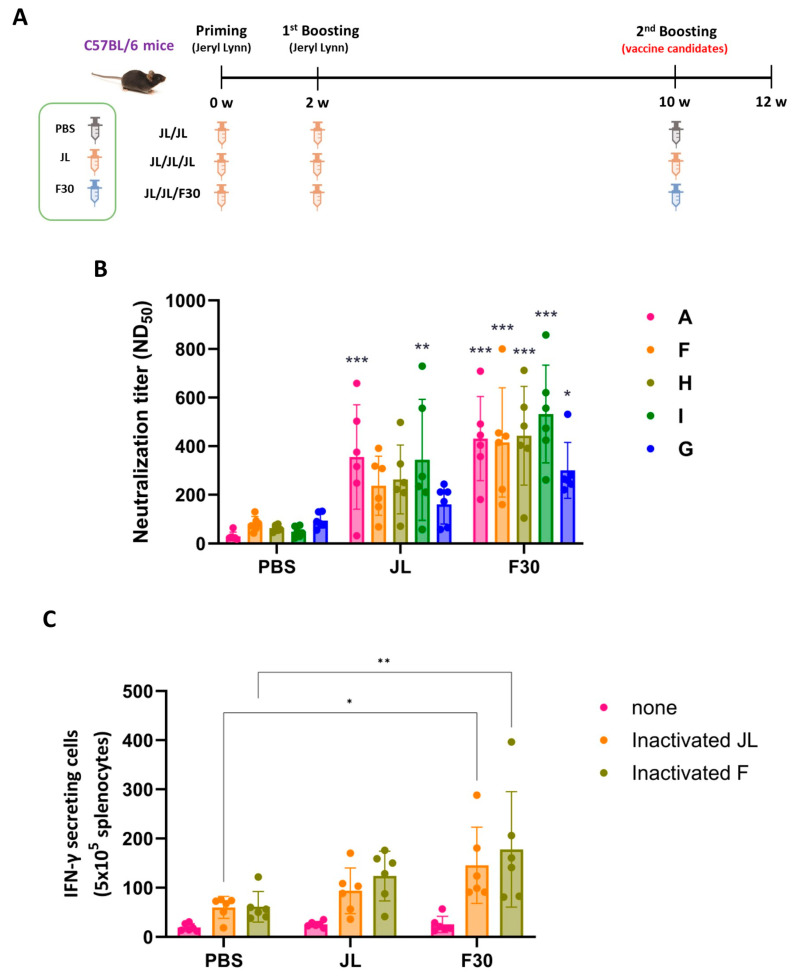
Immunogenicity of F30 as a third dose immunization in C57BL/6 mice. (**A**) Experimental schematic of the three-dose immunization. Mice were immunized with the prime, first booster of JL, and second booster of PBS, JL, or F30 (*n* = 6). (**B**) Neutralizing antibody titers toward the indicated MuV genotypes determined by PRNT assays. Two-way ANOVA and Tukey’s multiple comparison test were performed; * *p* < 0.05, ** *p* < 0.01, *** *p* < 0.001 vs. PBS group. (**C**) ELISpot assay of inactivated MuV antigen-specific IFN-γ secretion from splenocytes from three-dose immunized mice. Splenocytes from immunized mice were isolated and stimulated with inactivated F30 and JL viruses. Two-way ANOVA and Tukey’s multiple comparison test were performed; * *p* < 0.05, ** *p* < 0.01 vs. PBS group.

**Figure 5 vaccines-12-00595-f005:**
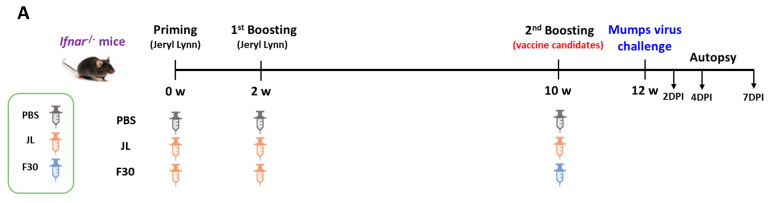
Cross-protection efficacy of F30-third-immunized *Ifnar*^−/−^ mice against circulating genotypes F and G. (**A**) Experimental model for third immunization with F30 and challenge with MuV. *Ifnar*^−/−^ mice were immunized with three doses of JL or two doses of JL and one dose of F30, and then challenged with 3 × 10^6^ PFUs of genotype F or G MuV (*n* = 2–5). Viral loads in the lungs from genotype F-challenged (**B**) and genotype G-challenged (**C**) mice measured by qRT-PCR. (**D**) Neutralizing antibody titers against the indicated MuV genotypes determined using PRNT assays with serum from the immunized *Ifnar*^−/−^ mice. Two-way ANOVA and Tukey’s multiple comparison test were performed; * *p* < 0.05, ** *p* < 0.01, *** *p* < 0.001 vs. PBS group, # *p* < 0.05, ### *p* < 0.001 vs. JL group.

**Table 1 vaccines-12-00595-t001:** Nucleotide and amino acid changes between mumps F genotype virus reference (F0) and cell culture-adapted viruses (F10 and F30).

Gene ID	Nucleotide Position	Nucleotide			Amino Acid Position	Amino Acid		
		F0	F10	F30		F0	F10	F30
-	11	-	A	G	-	-	-	-
-	14	T	T	A	-	-	-	-
NP	503	G	G	A	120	Ala	Ala	Thr
	1738	C	C	T	531	Thr	Thr	Thr
F	4777	G	A	A	78	Asp	Asn	Asn
	5350	A	G	G	269	Met	Val	Val
SH	6437	T	T	C	57	Leu	Leu	Pro
-	6586	A	T	T	-	-	-	-
HN	7073	A	A	G	154	Thr	Thr	Ala
	8105	C	A	A	498	His	Asn	Asn
L	9349	A	A	G	304	Gln	Gln	Gln
	10,889	C	C	A	818	His	His	Asn
	12,654	A	G	G	1406	Lys	Arg	Arg
	14,274	C	C	A	1946	Pro	Pro	Gln

Whole-genome sequencing revealed that the amino acid sequences were altered in the nucleoprotein (NP), fusion (F), small hydrophobic (SH), hemagglutinin-neuraminidase (HN), and large (L) proteins. The reference sequence (F0) was MuVi/Incheon.KOR/16.08/22 from GenBank no. MN630056.1. The bolds indicate amino acid sites that have changed in F30 sequence.

## Data Availability

The data that support the findings of this study are available from the corresponding author upon reasonable request.

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
