# Peer review of "Immunogenicity and Cross-Protection Efficacy of a Genotype F-Derived Attenuated Virus Vaccine Candidate against Mumps Virus in Mice"

_vaccines, 2024, doi:10.3390/vaccines12060595_

Round 1

Reviewer 1 Report (Previous Reviewer 1)

Comments and Suggestions for Authors

The revised manuscript has addressed my concerns.  I have no further concerns with this manuscript.

Reviewer 2 Report (Previous Reviewer 2)

Comments and Suggestions for Authors

Thank you for the new version of this manuscript. The authors have addressed all my comments appropriately in the revised version of the manuscript. 

Reviewer 3 Report (Previous Reviewer 3)

Comments and Suggestions for Authors

In this paper, Kim and colleagues report on the development of a cell-culture adapted, live attenuated genotype F mumps vaccine candidate (F30). They found that the F30 vaccine elicited cross-reactive antibody responses with other mumps genotypes, and the data presented suggest that the F30 vaccine may be potentially useful as a booster dose. The authors addressed one of my primary concerns from the previous review by including the JL vaccine strain as a comparator in the neurotoxicity studies. The authors have also considerably softened their stance on the proposed utility of the F30 vaccine as a booster vaccine, which the data suggests it may be beneficial but, in my opinion, this is not definitive evidence – there is clearly a benefit in some immunological readouts, but for others, the F30 elicits only marginal improvements (if any) over JL. I still feel that the experiments chosen to quantify vaccine efficacy against viral challenge could be improved by conducting passive transfer rather than immunizing an immunocompromised mouse model, but the authors have substantially improved the manuscript in all other areas based on my previous notes. I recommend this work for publication without additional edits.

This manuscript is a resubmission of an earlier submission. The following is a list of the peer review reports and author responses from that submission.

Round 1

Reviewer 1 Report

Comments and Suggestions for Authors

In this study, an attenuated mumps virus vaccine candidate was generated through serial passage of a F genotype strain.  The goal is to develop a live attenuated vaccine that is a better match for circulating mumps viruses than Jeryl Lynn (JL). The passaged virus, termed F30, could elicit higher neutralizing titers in mice than JL against genotype F, H, I, and G viruses, and this was true even for mice primed with JL and then boosted with F30.  Overall these results are clear and the manuscript is well written.  My concerns are relatively minor.

1.  Results of Fig. 1 are difficult to interpret.  Why would adaptation of virus through repeated passage in Vero cells result in a modified virus that replicates worse in Vero cells compared to the starting virus?  Please discuss.

2.  Please discuss strengths and limitations of the newborn mouse neurotoxicity model vs. alternative models for mumps neurotoxicity that have been established.

3.  How does neurotoxicity of F30 virus compare with the standard JL vaccine strain?

Author Response

1.  Results of Fig. 1 are difficult to interpret.  Why would adaptation of virus through repeated passage in Vero cells result in a modified virus that replicates worse in Vero cells compared to the starting virus? Please discuss.
(Response) As suggested, we have added the related contents to the Discussion in the revised manuscript on page 13, line 418 to 422. “Recombinant MuV(S79) carrying mutation in L protein leads to delayed viral growth in cell culture and formation of significantly smaller plaques compared to parental virus. Despite the attenuated viral growth, cotton rats vaccinated with these rMuV mutants produced high levels of NAbs and were completely protected against challenge with a genotype F MuV [7].” and page 13, line 426 to 428 “evidence suggests that the HN and L proteins are associated with neuroattenuation in newborn mice and exhibit attenuated growth and alterations in plaque morphology in Vero cell culture.” 
2.  Please discuss strengths and limitations of the newborn mouse neurotoxicity model vs. alternative models for mumps neurotoxicity that have been established.
(Response) We have added this information in the discussion section of the revised manuscript on page 12-13, line 392 to 404. “Ferrets and monkeys are susceptible to MuV infection [5, 24], However, their models are expensive and difficult to manage. Additionally, some vaccines that appeared neuroattenuated (weakened) in monkeys were later found to be neurovirulent (pathogenic to the nervous system) in humans. This highlights the limitations of using small numbers of monkeys in safety testing. Small animal models, such as rats and mice, offer a better alternative because they are easier to handle and breed. They also allow for larger study sizes, which facilitates obtaining statistically significant results in research. Studies have shown that newborn rats can be used to evaluate the neurotoxicity of mumps vaccines by assessing the severity of hydrocephalus (fluid buildup in the brain) following intracerebral (in the brain) immunization with the mumps virus [25, 26]. While adult mice are resistant to MuV infection, newborn mice infected with the virus develop lung lesions and show detectable viral titers in the brain [27-29].” 
3.  How does neurotoxicity of F30 virus compare with the standard JL vaccine strain?
(Response) To compare the neurotoxicity of F30 and JL, we have included data for JL in Figure 2 and added related content to the Discussion in the text as follows: “Therefore, we employed a newborn mouse model to evaluate the neurotoxicity of our vaccine candidate. Our findings indicate that the F30 strain exhibited lower neurovirulence in neonatal mice compared to the JL and pre-attenuated F10 strains. Notably, the neurotoxicity of the JL strain was comparable to that of the F10 strain.” (page 13 line 404 to 408).

Reviewer 2 Report

Comments and Suggestions for Authors

In the present manuscript, Dr Kim and colleagues prepared a novel mumps virus genotype F vaccine candidate by passaging this virus 30 times. Subsequently, the immunogenicity and cross protection was evaluated in multiple mouse models. Overall, the study is well designed, experiments were well performed and the manuscript was clearly written in my opinion.

I have a few comments:

Line 66. Additional studies could be included as a reference

Paragraph 2.5 Please describe animal experiments more clearly, e.g anesthesia, euthanasia.

Paragraph 2.6. Please include group size

Line 425. Multiple additional steps are needed to bring this vaccine candidate into use for humans

Author Response

1. Line 66. Additional studies could be included as a reference
(Response) There are many reports that the shift away from genotype A strain has brought into question the efficacy of the Jeryl Lynn vaccine recently. Studies have been conducted to develop new vaccine candidates based on genotypes that are mainly prevalent depending on the region, and these studies have been added to the Introduction (page 2 line 72).
2. Paragraph 2.5 Please describe animal experiments more clearly, e.g anesthesia, euthanasia.
(Response) Twenty-five days after vaccination, the animals were anesthetized by administering appropriate doses of Rompun and Ketamine, and after sample collection, they were euthanized with CO2 gas according to procedures approved by KCDC's IACUC. We added the related contents to the Materials and Methods (page 3 line 134 to 135) “the mice were anesthetized by appropriate doses of Rompun and Ketamine, and euthanized with CO2 gas after sample collection as recommended.” and you can see the details about animal experimental approval on page 14 line 475-479.
3. Paragraph 2.6. Please include group size
(Response) Most animal experiments were conducted with 6 mice per group, and the number of mice was changed slightly depending on the situation. To reduce unnecessary euthanasia, the minimum number of mice was used for statistical analysis. The group size for each experiment is written in each figure legend.
4. Line 425. Multiple additional steps are needed to bring this vaccine candidate into use for humans
(Response) We agree with reviewer’s comment. Our study was an early stage of vaccine development to identify a potential vaccine candidate. We generated an attenuated mumps virus, F30 by serial passage in Vero cells and found the F30-third vaccinated mice produced higher neutralizing antibody against circulating mumps viruses than the JL-third immunized mice. We modified the related content to the Discussion in the text; “Based on these results, we hypothesize that a third dose, later in life, with a genotype F-derived F30 vaccine may be beneficial in preventing further resurgence of mumps.” (page 14 line 461 to 463). 

Reviewer 3 Report

Comments and Suggestions for Authors

In this paper, Kim and colleagues report on the development of a cell-culture adapted, live attenuated genotype F mumps vaccine candidate (F30). They found that the F30 vaccine elicited cross-reactive antibody responses with other mumps genotypes, but in my opinion, the data presented do not resoundingly support the use of the F30 vaccine as a booster dose as the authors claim. There are two main areas of concern – the first is the evaluation of neurotoxicity by the F30 vaccine strain. This is not directly compared to the current vaccine strain (JL), and I feel strongly this data should be included as the F30 vaccine does show some inflammation in brain tissue. Previous mumps vaccines have been licensed for use without proper consideration of this, leading to multiple cases of vaccine-induced meningitis. The second point of concern is with the mouse data, where the authors make several claims that the data does not support. Administering the F30 vaccine as a third dose elicits slightly higher neutralizing antibody titers and T cell responses compared to a third dose of JL; however, neither of these responses is statistically significant from the JL response. This is counterintuitive to the authors’ claims that the F30 vaccine should be considered for a booster dose later in life as it elicits greater neutralizing antibody responses and T cell responses. Larger numbers of mice included in the study may help to control the variability in the data and allow for statistically significant comparisons. The F30 vaccine also fails to demonstrate superior protection against viral challenge compared to JL – this experiment could be improved by passive transfer experiments (see notes below). The authors must significantly revise portions of the paper as noted below (including potential additional experiments) before this manuscript can be published.       

-       Line 49-74: The way the information is presented here is a bit confusing. It begins by stating that waning immunity is a possible explanation for the resurgence of mumps cases, but then pivots to discuss the historical epidemiology of mumps as a childhood disease in Korea before switching back to discussing how older age groups now have lower seroprevalence. Then, the concept of antigenic difference between genotypes is introduced. I think this could be better presented by first discussing disease epidemiology in Korea with low seroprevalence in older age groups, then introducing waning immunity and antigenic mismatch as potential hypotheses for the resurgence of mumps.  

-       Line 60-61: There is significant evidence emerging that additional doses of MMR do not offer any significant long-term boost in immunity against mumps.

-       Line 149-150: What volume of liquid was used for virus inoculation?

-       Section 2.6.1 ELISA: This explains how the plates were prepared and incubated after serum was added, but it does not explain what dilution(s) the serum was added to the plate, if samples were tested in singlet/duplicate/etc. These details should be included.

-       It would have been potentially useful to assess memory B cell ELISPOT responses in these animals, as B cell memory is likely an important contributor to protective immunity against mumps.

-       Line 226-227: I don’t know that I would say the growth was delayed, as the F30 virus appears to be growing at the same rate, just the virus titer for F30 never reaches that of JL or F10. It might be better to simply say that peak titers were lower.  

-       Line 234-235: Was the F30 plaque purified to ensure virus progeny were from a single clone? The irregularity in plaque size may suggest quasi-species or a mixture of viruses with different mutations.

-       Line 259-260: What would be considered safe for a mumps vaccine regarding inflammation score in the brain? This has been a problem with mumps vaccines that were under-attenuated in the past. The JL vaccine strain should be included in this experiment as a control and reference point for the reader.

-       Figure 2 caption: States that images are for mice inoculated with JL, F10, and F30 viruses but the image and manuscript text reference PBS instead of JL. Please correct.

-       Line 273: States that serum dilution yielding an OD 0.5 was used to determine titer, but please clarify how this was calculated (e.g., curve-fitting, etc.)

-       Figure 3D: A couple of points here. 1) why are mice immunized with PBS registering mumps titers? There shouldn’t be any mumps antibodies in this treatment group. Is this an artifact of how the data was calculated with the Karber formula?

-       Figure 4B: It is unclear what groups are being compared for the statistical significance values indicated. Please clearly indicate comparisons in the figure legend. Based on the spread and variability in the data, it seems the titers elicited by F30 and JL are statistically equivalent and these values are comparing to the PBS group.

-       Figure 4C: The data presented here does not justify the statement on Lines 309-310 that the F30 vaccine elicited a greater T cell response than JL. The error bars on these data overlap significantly and there is no statistical significance demonstrated. The statement on Lines 309-310 should be revised to soften this claim.

-       Section 3.6 and Figure 5: This experiment would have been better conducted as a passive transfer of serum from immunocompetent mice immunized with the vaccines of interest. The data presented in Figure 5 do not indicate protective efficacy despite significant differences from the PBS group – protective efficacy would have been no detection of mumps RNA in the lungs of vaccinated mice. Pickar et al demonstrate that mumps in the lung is virtually undetectable after passive transfer from immunized mice. The authors reference this in their discussion – I see no reason why the experiment cannot be performed and included here. Further, in the data presented, there appears to be no statistical difference between the JL and F30 vaccines in this experiment.

-       Line 425-426: I do not think the data in this paper definitively support that an F30 vaccine booster later in life will be beneficial in preventing mumps.

Author Response

1. Line 49-74: The way the information is presented here is a bit confusing. It begins by stating that waning immunity is a possible explanation for the resurgence of mumps cases, but then pivots to discuss the historical epidemiology of mumps as a childhood disease in Korea before switching back to discussing how older age groups now have lower seroprevalence. Then, the concept of antigenic difference between genotypes is introduced. I think this could be better presented by first discussing disease epidemiology in Korea with low seroprevalence in older age groups, then introducing waning immunity and antigenic mismatch as potential hypotheses for the resurgence of mumps.  
(Response) We have implemented the suggestion; we have rearranged and added a new information to the Introduction on page 1-2, line 36-66. “MuV is also highly neurotropic, with approximately half of all clinical cases demonstrating its ability to invade the central nervous system (CNS). It leads to meningitis in 10% of cases and encephalitis in less than 1% [4-7].
Live-attenuated vaccines have proven successful in combating various infectious viruses, such as smallpox, poliovirus, yellow fever, and the measles, mumps, and rubella (MMR) viruses [8]. These vaccines offer several advantages, including a relatively simple manufacturing process, preservation of native antigens, emulation of natural infections, and eliciting a robust immune response [9]. The first live-attenuated mumps vaccine, Jeryl Lynn (JL) B strain, belonging to genotype A, received approval in the USA in 1967. Subsequently, the mumps vaccine became part of the trivalent MMR vaccine. Implementation of the two-dose MMR vaccination program containing the JL strain significantly reduced mumps incidence among school children by the 1990s. Since the introduction of the two-dose MMR vaccine in the Republic of Korea in 1997, there has been a rapid decrease in mumps infections [1, 10]. 
However, despite vaccination efforts, significant mumps outbreaks continue to affect adolescents globally. According to mumps incidence data sourced from the Korea Disease Control and Prevention Agency’s infectious disease (https://dportal.kdca.go.kr/pot) spanning the decade from 2012 to 2022, over 15,000 mumps cases have been consistently reported annually since 2013. Notably, in 2014, there was a significant surge in mumps infections among adolescents aged 13–18 years, with reported cases reaching 13,603.
The resurgence of mumps outbreaks can be attributed to waning immunity and antigenic differences between vaccine strains and circulating strains. Findings from the "2014 Measles, Mumps, and Rubella Immunity Survey" conducted domestically revealed that the antibody positivity rate was 74% among individuals aged 2–3 years, 86% among those aged 4–6 years, and 89% among those aged 7–9 years. However, this rate decreased to 62% among individuals aged 13–18 years. Additionally, immunity surveys conducted overseas have reported a significant decline in antibody levels and disparities in the avidity of mumps immunoglobulin G (IgG) over time post-vaccination, compared to measles and rubella [11, 12]. Therefore, waning immunity post-vaccination emerges as a plausible explanation for the increased incidence of mumps despite high vaccination rates. Hence, additional vaccination is recommended for children and adolescents.”
2. Line 60-61: There is significant evidence emerging that additional doses of MMR do not offer any significant long-term boost in immunity against mumps.
(Response) Mumps has shown a resurgence during childhood or adolescence, even within highly vaccinated populations. This resurgence in mumps outbreaks can be attributed to waning immunity and antigenic differences between vaccine strains and circulating strains. The related information has been added to the Introduction on page 2, line 56 to 72. “The resurgence of mumps outbreaks can be attributed to waning immunity and antigenic differences between vaccine strains and circulating strains. Findings from the "2014 Measles, Mumps, and Rubella Immunity Survey" conducted domestically revealed that the antibody positivity rate was 74% among individuals aged 2–3 years, 86% among those aged 4–6 years, and 89% among those aged 7–9 years. However, this rate decreased to 62% among individuals aged 13–18 years. Additionally, immunity surveys conducted overseas have reported a significant decline in antibody levels and disparities in the avidity of mumps immunoglobulin G (IgG) over time post-vaccination, compared to measles and rubella [11, 12]. Therefore, waning immunity post-vaccination emerges as a plausible explanation for the increased incidence of mumps despite high vaccination rates. Hence, additional vaccination is recommended for children and adolescents.
The F, H, and I MuV genotypes are prevalent in East Asian countries, and the G genotype is prevalent in Europe and the United States [13, 14]. In Korea, the F genotype was briefly prevalent in 2008, and the H and I genotypes have been largely prevalent since 2007. It has been reported that the commercial genotype A vaccine strains (JL and RIT 4385) are genetically different from epidemic strains and that there may also be differences in immunogenicity [9, 10, 15-17].”
3. Line 149-150: What volume of liquid was used for virus inoculation?
(Response) We have added the related information to the Materials and Methods on page 4, line 153 to 156).
4. Section 2.6.1 ELISA: This explains how the plates were prepared and incubated after serum was added, but it does not explain what dilution(s) the serum was added to the plate, if samples were tested in singlet/duplicate/etc. These details should be included.
(Response) We have added the related information to the Materials and Methods in the revised manuscript on page 4, line 162-164. “The plates were washed and blocked with 200 µL of PBS containing 5% skim milk for 1 h at 25 °C. Serum was serially diluted (three-fold) starting at 1:100 in PBS containing 3% skim milk and added to each virus-coated well.”
5. It would have been potentially useful to assess memory B cell ELISPOT responses in these animals, as B cell memory is likely an important contributor to protective immunity against mumps.
(Response) To assess vaccine efficacy, the primary method involves measuring neutralizing antibody titers. Additionally, evaluating T-cell function has proven valuable in this regard. The T cell ELISPOT assay allows for the measurement of antigen-specific T cell responses within a relatively short timeframe. Therefore, in this study, we utilized PRNT assays and T cell ELISPOT assays to evaluate the vaccine efficacy of F30, as suggested by the reviewer. Furthermore, as per the reviewer's comment, integrating the B cell ELISPOT assay into our research can provide insights that may lead to the development of more effective vaccines capable of inducing protective immunity against various infectious diseases. In future follow-up studies, we intend to incorporate the B cell ELISPOT assay alongside the T cell ELISPOT assay, following your guidance.
6. Line 226-227: I don’t know that I would say the growth was delayed, as the F30 virus appears to be growing at the same rate, just the virus titer for F30 never reaches that of JL or F10. It might be better to simply say that peak titers were lower.  
(Response) As suggested, we have changed the related content in the Results as follows. “The virus growth of F30 was delayed by 1 day compared with the JL and F10. The peak titer of F30 was 6.32 log10 PFU/mL, which was lower than that of JL and F10 (7.46 log10 PFU/mL and 7.30 log10 PFU/mL) at 3 days after infection.” (page 6 line 231 to 234)
7. Line 234-235: Was the F30 plaque purified to ensure virus progeny were from a single clone? The irregularity in plaque size may suggest quasi-species or a mixture of viruses with different mutations.
(Response) Although we did not purify a single plaque, the peaks on sequencing electropherogram were almost homozygous except at the mutated sties. Therefore, we think the F-derived F30 is a single colony.
8. Line 259-260: What would be considered safe for a mumps vaccine regarding inflammation score in the brain? This has been a problem with mumps vaccines that were under-attenuated in the past. The JL vaccine strain should be included in this experiment as a control and reference point for the reader.
(Response) As noted in our response to reviewer 1's third point, we investigated the neurotoxicity of both F30 and JL. We have included data for JL in Figure 2 and expanded the discussion in the text to address this comparison. Specifically, in the Discussion section (page 13, lines 404 to 408), we have added the following statement: “Therefore, we employed a newborn mouse model to evaluate the neurotoxicity of our vaccine candidate. Our findings indicate that the F30 strain exhibited lower neurovirulence in neonatal mice compared to the JL and pre-attenuated F10 strains. Notably, the neurotoxicity of the JL strain was comparable to that of the F10 strain.” 
9. Figure 2 caption: States that images are for mice inoculated with JL, F10, and F30 viruses but the image and manuscript text reference PBS instead of JL. Please correct.
(Response) As suggested, we have correctly modified the legend in Figure 2 on page 7, line 268 to 273.
10. Line 273: States that serum dilution yielding an OD 0.5 was used to determine titer, but please clarify how this was calculated (e.g., curve-fitting, etc.)
(Response) We have added the related contents to the Materials and Methods and Results in the text; “the dilution factor at the cutoff value was calculated as the ELISA titer. The cutoff value was calculated by the mean absorbance at 450 nm of the negative control.” (page 4 line 168 to 170) and “OD 0.5, which was the mean absorbance at 450 nm of the PBS-immunized samples,” (page 7 line 280-281).
11. Figure 3D: A couple of points here. 1) why are mice immunized with PBS registering mumps titers? There shouldn’t be any mumps antibodies in this treatment group. Is this an artifact of how the data was calculated with the Karber formula?
(Response) The PBS-immunized mice group was used as a negative control and NAb titers of control sera from mice immunized with PBS were artifacts of the Karber formula with background inhibition levels. We have incorporated relevant information into the Results section of the manuscript as follows: “C57BL/6 mice were immunized twice with F30 or JL as a positive control at 2-week intervals and serum was collected at termination (Figure 3A). Another group of mice were immunized with PBS as a negative control.” (page 7 line 276 to 278) and “NAb titers of the control sera from the PBS-immunized mice yielded background levels of inhibition (Figure 3D).” (page 8 line 288 to 289).
12. Figure 4B: It is unclear what groups are being compared for the statistical significance values indicated. Please clearly indicate comparisons in the figure legend. Based on the spread and variability in the data, it seems the titers elicited by F30 and JL are statistically equivalent and these values are comparing to the PBS group.
(Response) As suggested, we have modified the legend in Figure 4 (page 10 line 322 to 330). “Immunogenicity of F30 as a third dose immunization in C57BL/6 mice. (A) Experimental schematic of the three-dose immunization. Mice were immunized with the prime, first booster of JL, and second booster of PBS, JL, or F30 (n = 6). (B) Neutralizing antibody titers toward the indicated MuV genotypes determined by PRNT assays. Two-way ANOVA and Tukey’s multiple comparison test were performed; *p < 0.05, **p < 0.01, ***p < 0.001 vs. PBS group. (C) ELISpot assay of inactivated MuV antigen-specific IFN-γ secretion from splenocytes from three-dose immunized mice. Splenocytes from immunized mice were isolated and stimulated with inactivated F30 and JL viruses. Two-way ANOVA and Tukey’s multiple comparison test were performed; *p < 0.05, **p < 0.01 vs. PBS group.”
13. Figure 4C: The data presented here does not justify the statement on Lines 309-310 that the F30 vaccine elicited a greater T cell response than JL. The error bars on these data overlap significantly and there is no statistical significance demonstrated. The statement on Lines 309-310 should be revised to soften this claim.
(Response) As suggested, we have changed the related contents to the Results in the revised manuscript as follows: “In an experiment to examine the T cell immune response through IFN-γ ELISpot assays, the IFN-γ levels were comparable between the F30-third immunized group and the JL-third immunized mice stimulated with both inactivated MuV JL and genotype F (Figure 4C).” (page 9 line 317 to 320).
14. Section 3.6 and Figure 5: This experiment would have been better conducted as a passive transfer of serum from immunocompetent mice immunized with the vaccines of interest. The data presented in Figure 5 do not indicate protective efficacy despite significant differences from the PBS group – protective efficacy would have been no detection of mumps RNA in the lungs of vaccinated mice. Pickar et al demonstrate that mumps in the lung is virtually undetectable after passive transfer from immunized mice. The authors reference this in their discussion – I see no reason why the experiment cannot be performed and included here. Further, in the data presented, there appears to be no statistical difference between the JL and F30 vaccines in this experiment.
(Response) We found that there was no difference the levels of mumps viral genomes in lung tissues in the F30- and JL-third vaccinated Ifnar-/- mice. It may be the lack of a complete immune response in Ifnar-/- mice. Passive transfer of serum from naïve mice after the third immunization with F30 to Ifnar-/- mice is good method to test of efficacy of F30 as a booster vaccine. However, due to practical constraints, including the need for a new Institutional Animal Care and Use Committee (IACUC) review and the waiting period required for acquiring Ifnar-/- mice from overseas, we were unable to conduct the suggested experiments within the designated timeframe. Instead, we opted to perform supplementary experiments to validate cross-neutralizing antibody titers in serum from immunized Ifnar-/- mice. We have included the new data in Figure 5D (page 12, lines 359 to 363) “We compared the production of NAbs after the third vaccination with JL and F30. NAb titers in the F30-third immunized group against genotypes A and I were akin to those in the JL-third immunized group. However, the F30-third vaccinated group exhibited higher antibody titers against circulating genotypes F, H, and G compared to the JL-immunized group. While both F30- and JL-third vaccination groups demonstrated similar protective effects against genotypes F and G in Ifnar-/- mice, these findings suggest that F30 could serve as an alternative booster vaccine candidate to curb the resurgence of mumps, considering the prevalence of current circulating genotypes.” (page 11 line 345 to 353).
15. Line 425-426: I do not think the data in this paper definitively support that an F30 vaccine booster later in life will be beneficial in preventing mumps.
(Response) While we did not explicitly demonstrate the protective efficacy of F30 as a third booster vaccine in Ifnar-/- mice, our findings indicate promising outcomes. Specifically, mice immunized with F30 as a third booster vaccine exhibited higher neutralizing antibody titers against circulating mumps viruses compared to those immunized with the JL vaccine strain. Furthermore, F30 displayed lower neurovirulence than the JL vaccine strain. Taken together, these results suggest that the attenuated F30 strain holds potential as a booster vaccine candidate. In light of these findings, we have revised the related content in the Discussion section of the text as follows: “Based on these results, we hypothesize that a third dose, later in life, with a genotype F-derived F30 vaccine may be beneficial in preventing further resurgence of mumps.” (page 14 line 461 to 463).
